# ROR: Nuclear Receptor for Melatonin or Not?

**DOI:** 10.3390/molecules26092693

**Published:** 2021-05-04

**Authors:** Haozhen Ma, Jun Kang, Wenguo Fan, Hongwen He, Fang Huang

**Affiliations:** 1Hospital of Stomatology, Guanghua School of Stomatology, Sun Yat-sen University, Guangzhou 510055, China; mahzh5@mail2.sysu.edu.cn (H.M.); kangj5@mail2.sysu.edu.cn (J.K.); fanweng@mail.sysu.edu.cn (W.F.); 2Guangdong Provincial Key Laboratory of Stomatology, Guangzhou 510080, China; 3Guanghua School of Stomatology, Sun Yat-sen University, Guangzhou 510080, China

**Keywords:** retinoic acid-related orphan receptor, melatonin, nuclear receptor, ligand

## Abstract

Whether the retinoic acid-related orphan receptor (ROR) is a nuclear receptor of melatonin remains controversial. ROR is inextricably linked to melatonin in terms of its expression, function, and mechanism of action. Additionally, studies have illustrated that melatonin functions analogous to ROR ligands, thereby modulating the transcriptional activity of ROR. However, studies supporting these interactions have since been withdrawn. Furthermore, recent crystallographic evidence does not support the view that ROR is a nuclear receptor of melatonin. Some other studies have proposed that melatonin indirectly regulates ROR activity rather than directly binding to ROR. This review aims to delve into the complex relationship of the ROR receptor with melatonin in terms of its structure, expression, function, and mechanism. Thus, we provide the latest evidence and views on direct binding as well as indirect regulation of ROR by melatonin, dissecting both viewpoints in-depth to provide a more comprehensive perspective on this issue.

## 1. Introduction

Nuclear receptors with typical domain structures and conserved sequences, such as thyroid hormone and sterol receptors, are a superfamily of ligand-dependent transcription factors [1,2]. In the 1980s, as new members of this superfamily were being explored, a number of orphan receptors with unknown ligands were identified, including retinoic acid-related orphan receptor (ROR) [3].

As a ligand-dependent transcription factor, ROR modulates the transcription of target genes by binding to ROR response elements (RORE) present in the target genes. ROR is implicated in modulating a variety of physiological processes, including cerebellar development, lymphoid tissue development, retinal development, bone formation, lipid metabolism, circadian rhythm, oxidative stress, and inflammation suppression. Furthermore, ROR is also a promising therapeutic target for autoimmune diseases, tumours, obesity, diabetes, and other diseases. Thus, the identification of novel natural and synthetic ROR ligands is a research hotspot, and further studies are required to facilitate the establishment of targeted therapies for the aforementioned physiological processes and diseases [4]. 

Melatonin (*N*-acetyl-5-methoxytryptamine) is an evolutionary molecule found in bacteria, fungi, algae, plants, invertebrates, and vertebrates. It regulates multiple functions that have developed along the evolutionary timescale. In humans, melatonin is pleiotropic [5]. Melatonin modulates sleep-wake rhythms, reproduction, and bone metabolism; it also affects the immune, neurological, cardiovascular, digestive, urinary, and endocrine systems. Moreover, it exhibits anti-inflammatory, antioxidant, anti-infective, and anti-tumour properties while also being a potential therapeutic agent for obesity, cardiovascular diseases, and neurodegenerative disorders. The functions of melatonin are often mediated through its binding to either type 1 or type 2 melatonin membrane-bound receptors (MT1/MT2) [6]. Activation of these different membrane receptors may lead to opposing outcomes. For instance, the MT1-mediated vasoconstrictive effects of melatonin are in contrast to the MT2-mediated vasodilatory effects [7]. Melatonin also functions through non-receptor-mediated mechanisms, such as its powerful free radical scavenging capability [6]. 

Melatonin is speculated to have nuclear binding sites, in addition to its membrane-bound receptors. In the 1990s, studies reported melatonin binding sites in purified cell nuclei derived from rat organs, including the liver, spleen, and thymus [8,9]. Contemporaneous studies illustrated that melatonin bound and activated the nuclear receptors RORα and RORβ, bringing the claim that ROR was the nuclear receptor of melatonin to a culmination [10,11]. This claim was partially corroborated by subsequent studies where several functions of melatonin were determined to be mediated by ROR. Furthermore, melatonin was found to play a ROR-ligand-like role [12,13]. Another study reported the co-localization and co-immunoprecipitation of melatonin and ROR, confirming their interaction [14]. However, this claim has faced scepticism and the study asserting that the direct binding between retinoid Z receptors β (RZRβ, now known as RORβ) and melatonin was retracted as it could not be reproduced [10]. Since then, several studies have demonstrated that melatonin indirectly regulates the transcriptional activity of ROR through intermediate steps [15,16]. Crystallographic evidence suggests that melatonin and its metabolites are not high-affinity ligands for ROR [17]. Therefore, a debate is ongoing on whether ROR is a nuclear receptor of melatonin. 

This review discusses the complex relationships of ROR with melatonin in terms of ROR’s structure, expression, function, and mechanism. We provide the latest evidence on both direct binding or indirect regulation of ROR by melatonin, and dissect both aspects in-depth in an attempt to provide a more comprehensive perspective on this issue. 

## 2. Expression and Functions of Retinoic Acid-Related Orphan Receptor

### 2.1. Basic Information of Retinoic Acid-Related Orphan Receptor 

In 1993, melatonin was detected in the nuclei of cells, which led to the inference of the presence of melatonin’s nuclear receptors [9,18,19]. Subsequently, it was proposed that the ROR family comprised the nuclear receptors for melatonin, with RORα1 and RORα2 (splicing variants of RORα), RZRβ (now known as RORβ) [10], and RORγ being the main binding sites.

The ROR group includes three members: RORα (RORA, NR1F1, or RZRα), RORβ (RORB, NR1F2, or RZRβ), and RORγ (RORC or NR1F3) [20]. Four RORα isoforms (RORα1–4) have been identified in humans, whereas only two have been reported in mice (RORα1 and RORα4). Murine expression of the RORβ1 and RORβ2 isoforms have been reported, but only the RORβ1 isoform has also been identified in humans. Both humans and mice express RORγ1 and RORγ2 (RORγ2 is also known as RORγt).

The expression patterns for the three members of the ROR group varies, and some isoforms appear to be tissue-specific. RORα is expressed in a broad range of tissues and organs, including the skin, adipose tissue, skeletal muscles, cerebellum, liver, kidneys, and thymus. In mice, high expression levels of RORα1 and RORα4 are reported in the cerebellum; however, in other tissues, RORα4 is the predominantly expressed form. Human RORα3, thus far, has been detected only in the testes. RORβ expression is restricted to the central nervous system, where it plays a role in sensory and circadian rhythms, as well as functions within the spinal cord and sensory organs [21]. RORβ1 has been detected in the cerebral cortex, pituitary gland, spinal cord, cochlea, and retina, while RORβ2 has been identified only in the rod photoreceptors and the pineal gland [22,23]. RORγ1 expression is prevalent in the muscles, adipose tissue, liver, and kidneys, similar to RORα [24]. However, RORγt, also known as the thymic isoform, is confined to certain immune cells, such as group 3 innate lymphoid cells (ILC3) [25] and T helper 17 (Th17) cells [26].

### 2.2. Functions of Retinoic Acid-Related Orphan Receptor: Related to or Independent of Melatonin

The RORs mainly regulate physiological and pathological processes, including immunity, development (especially the nervous system), circadian rhythm, tumours, cellular metabolism, and oxidative stress, which is also the main functional areas of melatonin. Moreover, various studies have reported that RORs mediate functions of melatonin [27,28,29,30] and that melatonin can regulate the transcriptional activity of RORs, boosting the transcription of ROR target genes [12,13,14]. However, RORs have also been found to function independently of melatonin-induced signalling [21,31,32,33,34].

RORα is intimately linked to the melatonin function and is extensively involved in the pleiotropic effects of melatonin [27,28,29,30,35,36,37,38,39,40]. The melatonin-RORα axis plays a role in immune, reproductive, and cardiovascular systems, as well as a variety of physiological and pathological processes such as oxidative stress, circadian rhythms, development, and oncogenesis. In the immune system, the melatonin-RORα pathway is crucial for the induction of T cell differentiation, downregulation of autoimmune responses, and anti-inflammatory functions. García et al. demonstrated that melatonin relies on RORα to suppress nuclear factor kappa-B (NF-κB)-related inflammatory responses; moreover, it promoted antioxidant defence during sepsis treatment [30]. Multiple sclerosis progression is tamed by restraining Th17 cell differentiation and stimulating type 1 regulatory T (Tr1) cell differentiation via the melatonin-RORα axis [28]. In systemic lupus erythematosus, another autoimmune disease, melatonin protects endothelial cells via a RORα-dependent mechanism by reducing the expression of inflammatory factors and inhibiting macrophage migration [29]. Melatonin protects vessels from rupture of hyperlipidaemia-induced arterial plaques via RORα-regulated macrophage polarisation [27]. Regarding the reproductive system, the melatonin-RORα pathway inhibits apoptosis of Leydig cells (testicular mesenchymal stromal cells) and promotes testosterone secretion [35,41]. In addition, melatonin ameliorates oxidative stress during cardiac hypertrophy and cardio/cerebral ischemia-reperfusion in a RORα-dependent manner, with beneficial effects on pathological cardiac hypertrophy, myocardial infarction, ischemic stroke, and diabetic cardiomyopathy [12,36,37,38]. For cancer suppression, RORα, as a circadian rhythm-related gene, contributes to melatonin-induced suppression of liver and colorectal cancers [42,43]. In addition, melatonin inhibits the activation of hepatic stellate cells through RORα-mediated suppression of 5-lipoxygenase [44]. RORα also mediates the role of melatonin in regulating hair growth and reducing neuropathic pain [45,46,47]. Both melatonin and RORα are associated with the regulation of bone and lipid metabolism, insulin secretion, and glucose tolerance; however, it is difficult to determine whether their effects are related [5,48,49]. 

However, several functions of RORα are irrelevant to melatonin signalling. For example, RORα is critical for the development of cerebellar Purkinje cells. As a result, mice lacking RORα show cerebellar atrophy and ataxia [31]. However, no implication of melatonin in cerebellar development has been reported.

RORβ and its functions are less relevant to melatonin signalling. RORβ is overexpressed at sites that regulate circadian rhythms, including the suprachiasmatic nuclei (SCN) and pineal gland. Fluctuations in the mRNA levels of RORβ in the pineal gland are similar to that of melatonin synthesis. However, Masana et al. showed that RORβ did not participate in melatonin-mediated circadian rhythm and phase shifting, despite the rhythmic nature of its expression [50]. Moreover, RORβ plays a vital role in retinal formation and function; it promotes the proliferation of retinal progenitor cells while also acting as a key transcription factor for modulating the differentiation of rod photoreceptors, amacrine cells, and horizontal cells [23,51]. Rods are photoreceptor cell that senses low light stimuli and are sensitive to light intensity. Amacrine and horizontal cells are responsible for the integration of visual signals. Furthermore, RORβ plays a crucial role in perceptual and cognitive functions by facilitating the formation of layered structures in the cerebral cortex, which are derived from the migration and differentiation of neurons. RORβ reportedly acts as a marker for neurons in layers IV and V of the cerebral cortex, as well as in the barrel cortex [21,32]. Mutations in the human RORβ gene contribute to epilepsy and intellectual deficiencies [33,34]. The absence of RORβ in mice also results in gait abnormalities and the suppression of reproductive functions [21]. However, no studies have reported the aforementioned functions for melatonin. Interestingly, recent in vitro and in vivo data has revealed that knocking down or genetically deleting RORβ boosts osteogenesis and restrains bone resorption; therefore, this could be applied as a new target for the treatment of osteoporosis [52,53]. Coincidentally, melatonin reportedly promotes osteogenic differentiation; thus, it can also be used to treat osteoporosis [54].

RORγ mediates several processes regulated by melatonin, with major reports indicating its involvement in the immune system and tumour suppression. Impaired thymopoiesis, deficiency of Peyer’s patches and peripheral lymph nodes, and a high incidence of thymic lymphoma have been reported in RORγ−/− mice [55,56]. Furthermore, RORγ has been identified as a key modulator of the differentiation and development of Th17 cells, ILC3 cells, and γδ T cells [25,57,58]. Moreover, by regulating these cells, RORγ has emerged as a potential therapeutic target for autoimmune diseases such as multiple sclerosis, asthma, lupus, atopic dermatitis, inflammatory bowel disease, arthritis, and psoriasis [59,60]. More importantly, melatonin can regulate the immune system via RORγ. Farez et al. demonstrated that melatonin contributes to the amelioration of autoimmune diseases, such as multiple sclerosis, by both suppressing the differentiation of pathogenic Th17 cells via RORγ, and promoting the development of protective Tr1 via RORα [61]. Xiong et al. revealed that the melatonin-RORα/γ pathway mediates the green light-induced proliferation of chicken T lymphocytes [62]. In addition to the immune system, the melatonin-RORγ axis also suppresses cancer [39]. In vitro and in vivo studies have shown that melatonin may induce growth and angiogenesis suppression in gastric cancer cells by downregulating RORγ expression [39]. RORγ has also been suggested as a possible emerging target for the treatment of obesity-related insulin resistance. Under high-fat diet conditions, RORγ-/- mice exhibit less fat accumulation and reduced insulin resistance [40]. Coincidentally, melatonin can also induce preadipocyte apoptosis, diminish visceral fat, and ameliorate insulin resistance [63,64]. Although both melatonin and RORγ are recognised as potential therapeutic agents for obesity and diabetes, the role of the melatonin-RORγ axis in adiposity and glucose metabolism has not yet been reported.

## 3. Mechanism of Action of Retinoic Acid-Related Orphan Receptor

### 3.1. Gene Transcription Modulation

With highly conserved sequences, the *ROR* genes encode proteins spanning 459–556 amino acids. Similar to other nuclear receptors, ROR has a classical structure consisting of four functional domains, including an amino-terminal (A/B) domain, a conserved DNA-binding domain (DBD) with two zinc fingers, a ligand-binding domain (LBD) located at the carboxyl terminus, and a hinge [65]. The transcription factor ROR modulates target gene transcription by binding to RORE in the target gene (Figure 1). DBD identifies RORE, which is composed of the AGGTCA core motif and subsequent A/T-rich sequences. Specifically, the interaction between ROR and RORE is accomplished by identifying the core motif in the main groove by p-box, which is the ring between the last two cysteines in the first zinc finger. Furthermore, the C-terminal extension, a region downstream of the two zinc fingers that can identify the 5′-AT-rich RORE segment in the small adjoining groove, also contribute to the binding of ROR to RORE [66]. The multifunctional LBD mediates ligand binding, co-activator or co-repressor recruitment, and nuclear localisation [66]. Ligands binding triggers conformational alterations of the LBD, with the subsequent recruitment of co-activators or co-repressors, ultimately boosting or repressing the target gene transcription. Recruitment of co-activators induces chromatin remodelling through histone acetylase, which results in increased gene expression [67]. In contrast, co-repressors induce chromatin compaction via histone deacetylase, thereby inducing the repression of gene expression. The co-activators recruited by ROR receptors include NCOA1, NCOA2, NCOA3, NCOA6, HTATIP, PPARGC1A(PGC-1α), and CTNNB1 (β-catenin), while co-repressors include NCOR1, NCOR2, NRIP1 (RIP140), NIX1, and HR (Hairless) [68,69,70]. 

In addition to the cofactors, the crosstalk of nuclear receptors and the post-translational modifications of ROR also participate in modulating ROR activity. The competitive binding to RORE by ROR and REV-ERB is a typical example of crosstalk between nuclear receptors (Figure 1). ROR and REV-ERB are ligand-dependent transcription factors. Ligand binding triggers conformational alterations in both ROR or REV-ERB. As a result, they recruit co-activators or co-repressors to ultimately boost or repress target gene transcription. REV-ERB and ROR are often co-expressed, and both specifically recognise RORE. Consequently, ROR and REV-ERB competitively bind to RORE present in the regulatory region of target genes, forming a mutually antagonistic relationship. In most cases, ROR recruits co-activators and promotes gene transcription, whereas REV-ERB recruits co-repressors and inhibits gene transcription. For instance, brain and muscle Arnt-like-1(BMAL1) is a key molecule regulating the biological clock. Here, ROR and REV-ERB, which are expressed in an opposing circadian oscillatory manner, dynamically regulate the expression of the target gene BMAL1 by competitively binding to the RORE of BMAL1, thereby inducing contrasting effects. Through this regulation, the expression levels of BMAL1 also undergo circadian oscillations, which are vital for the proper functioning of circadian rhythms [71] (Figure 1). Ubiquitination and SUMOylation also modulate the transcriptional activity of ROR. As mentioned previously, RORγ alters IL-17 expression and consequently influences the development of inflammatory diseases. Recent reports indicate that the ubiquitination and SUMOylation of RORγ suppress RORγ, which subsequently represses colonic inflammation [72,73].

### 3.2. Ligands

ROR is dependent on ligands for transcriptional modulation. Such ligands are categorised by their impact on receptor transcriptional activity, comprising agonists, antagonists, and inverse agonists. Agonist binding induces increased co-activator recruitment and enhances the transcription of target genes. Antagonists have no influence on the transcriptional activity of the receptor but block agonist-mediated activation. However, inverse agonists recruit extra co-blockers to downregulate the basal activity of the receptor [4] (Figure 2). The mechanism underlying inverse agonists remains unclear and may be attributed to the hindered binding of the co-activators [74].

The ROR family acts as a vital regulator of circadian rhythm, immunity, development, oxidative stress, and cellular metabolism. Novel ligands of ROR have been enthusiastically identified and synthesised to establish positive modulators and therapeutics. For example, the novel endogenous ligand of RORα, maresin 1, has been identified as a therapeutic agent for non-alcoholic steatohepatitis by activating RORα to regulate hepatic macrophage polarity [75]. In addition, SR1001, a high-affinity ligand for RORα and RORγ, was synthesised to effectively alleviate autoimmune diseases and type 1 diabetes by targeting RORα and RORγ to restrain the function of Th17 cells [76,77]. New putative ligands for ROR have been identified using strategies such as the AlphaScreen assay, differential scanning fluorimetry, virtual screening, and cell-based reporter assays. Table 1 illustrates the natural and synthetic ligands of ROR as well as their applications.

Regarding melatonin, luciferase reporter gene assays have confirmed that melatonin increases the transcriptional activity of RORα [13,61]. Co-localization and co-immunoprecipitation of melatonin and RORα were reported by Lardone et al. [14], illustrating the interaction between melatonin and RORα.

### 3.3. Related Pathways

Melatonin and ROR are dependent on similar signalling pathways. ROR is a crucial regulator of inflammation and immunity and always acts via the NF-κB pathway. For example, RORα suppresses LPS-induced inflammation triggered by the SIRT1-NF-κB pathway [99]. Through extracellular signal-regulated kinase (ERK) and the NF-κB pathways, ROR ameliorates brain inflammation [100]. Moreover, the NF-κB pathway also mediates the protective effects of the melatonin-RORα axis against ischemia-reperfusion [36]. In addition to the NF-κB signalling pathway, the AMP-activated protein kinase α-signal transducer and activator of transcription (AMPKα-STAT) and the IL-6-STAT3 pathways also mediate the anti-inflammatory effect of ROR [27,101]. Recently, Han et al. identified a novel pathway, known as the MaR1/RORα/12-LOX loop, which plays a potential therapeutic role in non-alcoholic steatohepatitis [75]. 

Melatonin also modulates inflammation via the NF-κB pathway [102]. Additionally, the p38 mitogen-activated protein kinase (MAPK) pathway, transforming growth factor-β1 (TGF-β1)/Smad3 pathway, Toll-like receptor 4 (TLR4)/MyD88/NF-κB pathway, and Janus kinase 2 (JAK2)-STAT3 pathway also mediate the effects of melatonin on the immune system. For instance, melatonin influences T/B cell activation and adjusts immune homeostasis via the p38 MAPK and NF-κB pathways [103]. Melatonin has also been used for the treatment of LPS-induced testicular inflammation in sheep by repressing the p38 MAPK pathway [104]. Furthermore, it reportedly relies on the STAT3 pathway to attenuate cancers associated with inflammation [105]. In addition, melatonin ameliorates renal inflammation by targeting the TGF-β1/Smad3 signalling pathway [106]. In mice, melatonin inhibits microglial inflammation and improves cognitive impairment by downregulating the TLR4/MyD88/NF-κB signalling pathway [102]. Melatonin treatment also reduces myocardial inflammation by targeting the JAK2-STAT3 pathway [107]. Moreover, it also reduces neuroinflammation via the Sirt1/Nrf2/heme oxygenase-1 (HO-1) route [108]. 

Regarding anticancer effects, ROR is reportedly associated with the Wnt/β-catenin signalling pathway. The ROR-Wnt/β-catenin axis regulates the carcinogenesis and progression of liver, colon, prostate, and colorectal cancers, providing emerging therapeutic targets for these cancers [109,110,111,112]. Furthermore, the proliferation of oral squamous cell carcinoma cells can be restrained by inhibiting the RORα-p53 pathway [113]. The anticancer effects of melatonin are mediated not only by the Wnt/β-catenin pathway but also through a complex network of signalling pathways, mainly involving the phosphatidylinositol 3-kinase/AKT/mechanistic target of rapamycin (PI3K/AKT/mTOR), NF-κB, ERK, and cyclooxygenase-2/prostaglandin E2 (COX-2/PGE2) signalling pathways, responsible for the anti-proliferative and pro-apoptotic effects on cancer cells [114,115], and c-Jun N-terminal kinase (JNK), Akt-MAPKs, and NF-κB pathways, responsible for anti-metastatic effects [116,117]. 

## 4. Retinoic Acid-Related Orphan Receptor and Melatonin: Direct Binding or Indirect Modulation?

### 4.1. Retinoic Acid-Related Orphan Receptor and Melatonin: Inextricably Linked 

Nuclear receptors for melatonin are thought to exist because of melatonin’s ability to penetrate cells and the detection of melatonin’s nuclear binding site [8]. In 1994, Becker-André et al. revealed that melatonin bound to and activated the nuclear receptor RZRβ (now known as RORβ) [10]. One year later, Wiesenberg et al. reported that melatonin could also bind to and activate RORα; moreover, they identified a novel synthetic ligand of RORα, CGP 52608, which is similar to melatonin in function [11]. Besides, the binding of 2-[125I]iodomelatonin ([125I]melatonin) to purified cell nuclei in the spleen and thymus was demonstrated, and this binding of melatonin could be substituted by CGP 52608. The aforementioned studies implied that the nuclear receptor of melatonin might be ROR. Since this initial discovery, the belief that ROR serves as the nuclear receptor for melatonin has been extensively adopted by many researchers [14,28,38,118,119]. Furthermore, numerous studies have illustrated that ROR is linked to melatonin in a myriad of direct or indirect ways, supporting the proposition that melatonin is a ROR ligand (Table 2).

First, ROR appears to be related to melatonin in its expression pattern. Melatonin, which is secreted by the pineal gland at night, regulates circadian rhythm and the sleep and wake cycle. Coincidentally, some members of ROR are also expressed in the pineal gland and in other key sites that modulate circadian rhythms. In addition, all three members of ROR are expressed in a circadian oscillatory pattern [20]. Moreover, ROR is responsible for modulating circadian rhythms. ROR and REV-ERB form a mutually antagonistic relationship, whereby they competitively bind to RORE in the regulatory region of target genes. ROR promotes target gene transcription, whereas REV-ERB represses it. The expression of ROR and REV-ERB fluctuates circadian rhythmically with opposite trends and dynamically modulates the transcription of the target gene BMAL1. Thus, BMAL1, a key factor in the control of the mammalian circadian biological clock, is also expressed in a circadian pattern, resulting in the effective amplification of the circadian oscillation [71].

Functionally, many studies have suggested that melatonin and ROR have identical functions [5,40,48,52,54,63,64], and that ROR mediates a number of melatonin functions [27,28,29,30,35,36,37,38,39,40]. Moreover, melatonin is found to exert an additional effect on the transcription of target genes of ROR, as reported by studies covering a wide range of areas, including immune regulation, inflammation suppression, oxidative stress, circadian rhythms, tumour development, reproduction, hair growth, bone metabolism, and lipid metabolism. Both melatonin and ROR mediate bone and lipid metabolism. In addition, melatonin often functions through the mediation of ROR. As previously mentioned, the melatonin-ROR axis exerts therapeutic effects on systemic lupus erythematosus, hyperlipidemia, myocardial infarction, and ischemic stroke via suppressing inflammation and reducing oxidative stress. This axis also represses liver, gastric, and colorectal cancers, reduce neuropathic pain, and alleviate liver fibrosis. Furthermore, melatonin also modulates the transcriptional activity of ROR. For example, García et al. revealed that melatonin treatment remarkably elevated RORα expression while suppressing REV-ERB expression, contributing to the transcriptional upregulation of the ROR target gene, BMAL1. The activation of the melatonin-RORα pathway leads to NF-κB inhibition, resulting in an anti-inflammatory effect [30]. Using luciferase reporter gene assays and chromatin immunoprecipitation assays, Farez et al. demonstrated that melatonin promoted binding between ROR and the IL10 RORE, thus activating IL10 expression and boosting regulatory T cell differentiation to alleviate multiple sclerosis [61]. Zhao et al. found that RORα played a regulatory role in diabetic cardiomyopathy and that the addition of melatonin exerted an SR1078-like (RORα agonist) effect, stimulating the transcription of RORα target genes and ameliorating diabetic cardiomyopathy [12]. Xu et al. reported that in the presence of melatonin, RORα expression was upregulated, with a subsequent enhancement of RORα transactivation of manganese-dependent superoxide dismutase to resist pathological myocardial hypertrophy [37]. All this evidence supported that melatonin intensified ROR transcriptional activity and further boosted target gene transcription. 

More importantly, some studies have demonstrated that melatonin plays a role comparable to that of a determined ROR ligand. Kim et al. demonstrated that melatonin was characteristically similar to that of cholesterol sulphate (an agonist of RORα) and enhanced the transcriptional activity of the target gene hypoxia-inducible factor 1α [13]. Similarly, Zhao et al. reported that melatonin-induced target gene transcription was comparable to that in response to SR1078 (an agonist of RORα) [12]. Not only did melatonin trigger the transcriptional activation of ROR target genes in a manner that resembled ROR ligands, but it also co-localized and co-immunoprecipitated with ROR. Lardone et al. detected the co-localization of melatonin and ROR in the nucleus of Jurkat T cells, as well as their co-immunoprecipitation [14], providing evidence for the interaction between melatonin and ROR.

Currently, the recognised pathways through which melatonin exerts its biological effects include: (a) G protein-coupled membrane receptors (MT1, MT2) signalling, (b) putative nuclear receptors (RORα, β, γ) signalling, and (c) free radical elimination signalling unrelated to the receptors. In some cases, melatonin functions via ROR and not via free radical removal or by membrane receptors. For example, Huang et al. demonstrated that melatonin suppressed the proliferation of rat dental papilla cells and simultaneously boosted their differentiation, while luzindole (a competitive MT1/MT2 antagonist) failed to reverse the effects of melatonin, indicating that the aforementioned functions of melatonin are independent of membrane receptors [120]. Recently, they further demonstrated that ROR mediated the enhancement of dental papilla cell differentiation by melatonin [121]. It may be a piece of circumstantial evidence in support of ROR as a nuclear receptor of melatonin.

### 4.2. Retinoic Acid-Related Orphan Receptor and Melatonin: Indirect Modulation

In the late 1990s, the study claiming that melatonin bound to RORβ was withdrawn, as its findings were not reproducible [6,10]. Despite the retraction of the literature, unfortunately, many studies still adopted the theory that ROR served as melatonin’s nuclear receptor to explain the role of melatonin or ROR, and subsequent publications repeated these explanations. With the emergence of new evidence from crystallographic studies, a growing number of scholars have begun to oppose this claim, suggesting that melatonin may indirectly affect ROR activity via intermediate steps rather than directly binding to ROR. Therefore, whether ROR is a nuclear receptor for melatonin remains controversial (Table 2).

A study by Slominski et al. noted that melatonin and its metabolites, including 6OH-melatonin, AFMK (*N*^1^-acetyl-*N*^2^-formyl-5-methoxykynuramine), 5-methoxytriptophol, and 5-methoxytriptamine, exhibited no significant impact on the transcriptional activity of RORα and RORγ (*p* > 0.05) [17]. Molecular modelling further revealed that melatonin and its metabolites were not high-affinity ROR ligands [17].

Furthermore, some believe that melatonin cannot function as a ligand for ROR due to the significant differences in structure between melatonin and its metabolites and the identified ROR ligands (lipids, comprising cholesterol, steroids, and retinoids) [122]. However, it is worth noting that the argument of structural similarity between putative ligands should be made with caution, as the ligand-binding pockets of nuclear receptors are often highly plastic, allowing them to accommodate ligands with radically distinct structures. For instance, RXR is shown to bind tributyltin, retinoids, and fatty acids [1].

Although the direct binding between melatonin and ROR has been challenged, melatonin has been found to function via ROR or to modulates the transcriptional activity of ROR in a number of studies. To elaborate on this further, it has been proposed that melatonin indirectly regulates the transcriptional activity of ROR or modulates ROR expression and its downstream signalling pathways through certain steps. Recently, emerging studies have provided evidence for these proposals.

The first possibility is that melatonin directly binds to membrane receptors (MT1 or MT2), other receptors, or regulatory proteins (undefined), and further modulates the transcription or translation of ROR, contributing to alterations in ROR transcriptional activity. Some studies have validated this hypothesis. In MCF-7 breast cancer cells, both MT1 and RORα mediate the anti-tumour effects of melatonin. However, MT1 inhibitors reverse the melatonin-induced activation of ROR, suggesting that melatonin indirectly regulates RORα by the mediation of the MT1 receptor [15]. In addition, melatonin also acts against myocardial ischemia-reperfusion by directly binding to MT2 and subsequently regulating RORα signalling [16].

The second possibility is that melatonin acts via the melatonin-Sirt1-circadian oscillator pathway. Melatonin upregulates SIRT1 in multiple ways [123,124,125]. Furthermore, SIRT1, an NAD+-dependent protein deacetylase, moderates central and peripheral circadian oscillators [126]. The proposed mechanism postulates that SIRT1 deacetylates PPARγ coactivator-1α (PGC-1α) and the deacetylated form of PGC-1α binds to ROR, contributing to the transcriptional activation of ROR target genes. This implies that melatonin may moderate the transcriptional activity of ROR via downstream SIRT1 signalling [6].

A third explanation is that melatonin and its metabolites can alter cellular redox statuses (NADH/NAD and NADP/NADP ratios), which influence oxidative phosphorylation and the pentose phosphate pathways. This leads to adjustments in the circadian oscillator system, including the expression and the transcriptional activity of ROR. Another mechanism could be that melatonin induces alterations in mitochondrial activity, which in turn affects ROR activity [127].

Finally, it has been suggested that melatonin signalling directly regulates the transcriptional activity of the REV-ERB. Since REV-ERB competes with ROR for binding to RORE, REV-ERB regulation is equivalent to the reverse modulation on ROR transcriptional activity. Recent studies have provided evidence for this proposal [43].

Since Wiesenberg et al. reported that melatonin and the synthetic ligand CGP52608 similarly bound to and activated RORα, several studies have suggested that CGP52608 could mimic melatonin binding to the nuclear receptor ROR. However, this claim rests on the hypothesis that melatonin binds to ROR. If melatonin indirectly modulates ROR activity, then the melatonin-mimicking claims of CGP52608 are bound to collapse. Furthermore, melatonin acts differently from CGP52608. For instance, both melatonin and CGP52608 inhibit the growth of MCF-7 breast cancer cells, but CPG-52608 inhibits MCF-7 cell growth with a very different dose-response than melatonin [15]. Furthermore, melatonin boosts the proliferation of rat epididymal epithelial cells, while CGP52608 acts as an inhibitor [128]. Therefore, the use of CGP52608 as a mimic of melatonin for ROR binding assays should be discarded.

## 5. Perspectives

No consensus has been reached on whether ROR is a nuclear receptor of melatonin. Both the expression pattern and functions of ROR are linked to melatonin to some degree. Some studies have suggested that melatonin works akin to ROR ligands, regulating the transcriptional activity of ROR [12,13,14]. Co-localization and co-immunoprecipitation of melatonin and ROR have been reported, illustrating their interaction [14]. Nevertheless, another study claiming the binding and activation of RORβ by melatonin is unreproducible [10]. Furthermore, crystallographic research and molecular modelling have implied that melatonin and its metabolites are not likely to be ROR ligands [7]. Moreover, the existence of certain intermediate steps that allow melatonin to indirectly modulate ROR expression and function has been confirmed [15]. Other potential transitional steps and pathways have also been proposed.

Thus, further research is warranted to shed more light on this issue. First, there are few published studies that provide direct evidence for the binding between melatonin and ROR (such as co-immunoprecipitation assays). Replications conducted by different groups are needed to obtain more reliable and convincing results. Second, it is proposed that melatonin may be a mid-affinity ligand for ROR. However, high-affinity ligands for ROR, such as cholesterol and steroids, are commonly present in cell culture media of experimental systems. The impact of other ligands needs to be excluded when testing the binding of melatonin to ROR. Finally, some studies have concluded that melatonin performs certain functions by binding to the ROR, based solely on the exclusion of membrane receptor mediation. However, such conclusions are not rigorous, owing to the possible existence of other unidentified receptors or regulatory proteins for melatonin. Therefore, based on the available scientific evidence, we conclude that the mechanisms underlying the effects exerted by melatonin need to be explored further until reproducible and reliable results are made available. We believe that the reports on ROR and melatonin should be more cautious and conservative in their presentation till the final conclusions are drawn.

## Figures and Tables

**Figure 1 molecules-26-02693-f001:**
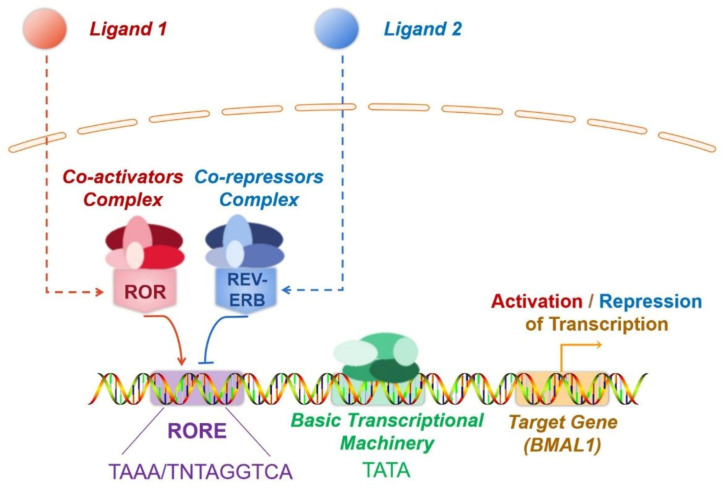
The mechanism of transcriptional modulation mechanism by ROR and REV-ERB. ROR and REV-ERB are ligand-dependent transcription factors. Ligand binding triggers conformational alterations in both ROR or REV-ERB. As a result, they recruit co-activators or co-repressors to ultimately boosts or represses target gene transcription. REV-ERB and ROR are often co-expressed, and both specifically recognise RORE. Consequently, ROR and REV-ERB competitively bind to RORE present in the regulatory region of target genes, forming a mutually antagonistic relationship. In most cases, ROR recruits co-activators and promotes gene transcription, whereas REV-ERB recruits co-repressors and inhibits gene transcription.

**Figure 2 molecules-26-02693-f002:**
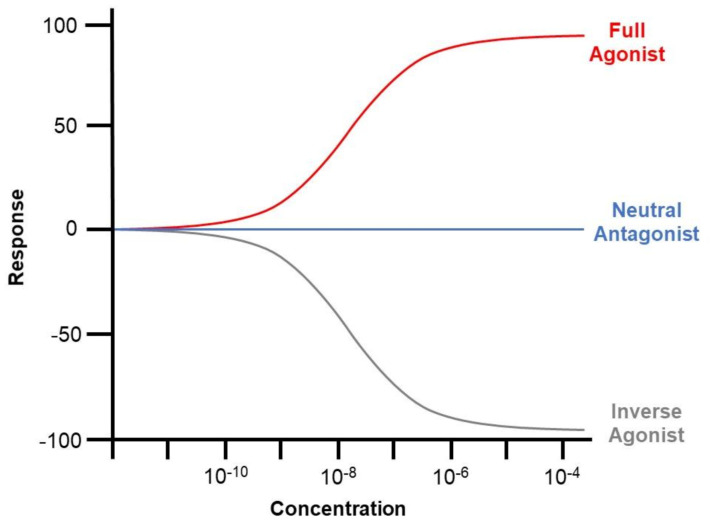
Graphical illustration of the impact of agonists, antagonists, and inverse agonists on the co-activator recruitment and downstream transcription of ROR. Agonist binding induces increased co-activator recruitment and enhances the transcription of target genes. Antagonists have no influence on the transcriptional activity of the receptor but block agonist-mediated activation. However, inverse agonists recruit extra co-blockers to downregulate the basal activity of the receptor.

**Table 1 molecules-26-02693-t001:** Ligands of retinoic acid-related orphan receptor.

Name	Type	Effect	Origin	Applications
**Natural Ligands**				
Cholesterol	RORα agonist	EC_50_: 200 nM ^1^	X-ray crystallography	Suppressing inflammation and dyslipidemia [78].
Cholesterol Sulfate	RORα agonist		X-ray crystallography	Regulating anticancer immunity [79] and ameliorating rheumatoid arthritis [80].
Stearic acid	RORβ antagonist		X-ray crystallography	Unclear
All-trans retinoic acid	RORβ antagonist	IC_50_ = 0.15 nM ^2^	X-ray crystallography	Unclear
Digoxin	RORγ inverse agonist (at high concentrations); RORγ agonist (at low concentrations)	IC_50_: 1.98 μM; EC_50_: unclear	Chemical screen	A therapeutic agent against Th17-mediated inflammation and autoimmune diseases, including encephalomyelitis [81], neovascular retinopathy [82], atherosclerosis [83] and autoimmune arthritis [84]; Inducing RORγ-dependent transcription in HepG2 and Th17 cells [85].
Ursolic Acid	RORγ inverse agonist	IC_50_: 680 nM	Chemical screen and luciferase reporter assays	A therapeutic agent against IL-17-mediated inflammation and autoimmune diseases [86].
7α-hydroxy-cholesterol	RORα/γ Inverse agonist	IC_50_: 1.3 μM (RORα); 0.62–2.2 μM (RORγ)	Chromatin immunoprecipitation	Unclear [87]
7β-hydroxy-cholesterol and 7-ketocholesterol	RORα/γ Inverse agonist	IC_50_: 0.7–1.4 μM (RORα); 0.62–2.2 μM (RORγ)	Chromatin immunoprecipitation	Unclear [87]
20α-hydroxycholesterol, 22R hydroxycholesterol and 25-hydroxycholesterol	RORγ agonist	EC_50_: 20–40 nM	Alphascreen	Unclear [88]
**Synthetic Ligands**				
ALTA 1550	RORβ antagonist	IC_50_ = 0.039 nM	Crystallography	Unclear
T0901317	RORα/γ inverse agonist	IC_50_: 2.0 μM (RORα); 1.7 μM (RORγ)	AlphaScreen	Unclear [89]
SR1001	RORα/γ inverse agonist	IC_50_: 172 nM (RORα); 117 nM (RORγ)	Luciferase reporter assays	A therapeutic agent against Th17-mediated diseases, including multiple sclerosis [76], type 1 diabetes [77], atopic dermatitis, and acute dermatitis [90].
SR1078	RORα/γ agonist	Unclear	Luciferase reporter assays	Regulating CD8 + T cell-mediated immune response [91], therapeutic strategy for rheumatoid arthritis [80] and acute kidney injury [92].
SR3335	RORα inverse agonist	IC_50_: 220 nM	Luciferase reporter assays	Slimming effect [93], suppression on elevated hepatic glucose production in type 2 diabetics [94], mitigation of sepsis stemming from *Enterococcus faecalis* oral infection [95].
SR2211	RORγ inverse agonist	IC_50_: 105 nM	Luciferase reporter assays	A therapeutic agent against Th17-mediated diseases [96] and doxorubicin resistance in prostate cancer [97].
SR1555	RORγ inverse agonist	IC_50_: 1.5 μM	Luciferase reporter assays	Anti-obesity [98] and therapeutic effects against Th17-mediated autoimmune diseases.

^1^ EC_50_: concentration for 50% of maximal effect. ^2^ IC_50_: half-maximal inhibitory concentration.

**Table 2 molecules-26-02693-t002:** Findings for or against the hypothesis of melatonin as a ROR ligand.

Findings	References
**In Support of Melatonin as a ROR Ligand**	
Melatonin binds and activates the nuclear receptor RZRβ.	[10]
Melatonin binds and activates RORα.	[11]
CGP 52608 (a novel synthetic ligand of RORα) is similar to melatonin in function. The binding of 2-[125I]iodomelatonin ([125I]melatonin) to purified cell nuclei in the spleen and thymus was demonstrated, and this binding of melatonin could be substituted by CGP 52608.	[11]
Melatonin and ROR have identical functions.	[5,40,48,52,54,63,64]
ROR mediates a number of melatonin functions.	[27,28,29,30,35,36,37,38,39,40]
Melatonin exerts an additional effect on the transcription of ROR target genes.	[12,30,37,61]
Co-localisation and co-immunoprecipitation of melatonin and ROR.	[14]
Melatonin functions via ROR without the mediation of membrane receptors.	[120,121]
**Refute that Melatonin is a ROR Ligand**	
The studies claimed that melatonin bound to RORβ was not reproducible.	[6,10]
ROR functions independently of melatonin-induced signalling.	[21,23,31,32,33,34,51]
Melatonin and its metabolites exhibit no significant impact on the transcriptional activity of RORα and RORγ.	[17]
Molecular modelling indicates that melatonin and its metabolites are not high-affinity ROR ligands.	[17]
Melatonin directly binds to membrane receptors (MT1 or MT2) and further modulates ROR.	[15,16]

## Data Availability

The data presented in this study are available on request from the corresponding author.

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
