# Peer review of "ROR: Nuclear Receptor for Melatonin or Not?"

_molecules, 2021, doi:10.3390/molecules26092693_

Round 1

Reviewer 1 Report

Dear Authors, I wish to compliment with your manuscript writing and simply recommend your review to be published without any scientific revision. Only a minor check on some English wording is required. The manuscript is well written providing an excellent overview of the historical and up-to-date state of the art on the molecular and functional cross-talk among melatonin and the different ROR molecules.

Reviewer 2 Report

In the present review Ma et al. present a comprehensive and detailed account of the still unresolved relationship between melatonin and the nuclear receptor ROR.  The review is well organized, generally clear and the arguments are adequately sustained.

I just have some points to raise.

Line 73: Remove “and”

Line 74: Replace “qualities” by perspectives/hypothesis, for instance

Lines 79, 319, 394: This statement is repeated several times. Please avoid repetitions throughout the text.

Line 110: “independently”

Line 111: Please provide reference(s) for this statement.

Line 225: “boost or repress” please correct

Line 235: “are vital” (the levels) please correct  

Line 264-266:   Some methodologies, such as cell-based reporter assays, do not measure direct binding but an effect produced upon exposure, hence the question on whether melatonin is able to bind or not RORs (the core of the present work); unless there are solid evidences for ROR-ligand interactions (binding assays, crystallography) can we affirm the compounds are true ligands? I would probably write “New putative/proposed ligands”

Table 1: ALTA1550 although retinoid-like is a synthetic compound, it should be placed with synthetic ligands.

Line 285: “identified”

Line 288: “modulates” instead of moderates

Line 327: Here the authors state that the hypothesis of melatonin as a ROR ligand has been widely proclaimed by many authors, which is true, then they go on pinpointing one specific scholar, Professor Russel J. Reiter, known for his works on melatonin and circadian rhythmicity.  I find this mention unnecessary. Also the following citations are from this author only. As the authors say, and I agree, this idea that melatonin is a bona fide ROR ligand is accepted by many.

Line 385: I would remove the titles, as they are not used in any other in text citations.

Line 404: The ligand binding pockets of nuclear receptors are often very plastic and able to accommodate ligands with radically distinct structures. For instance, RXR was shown to bind tributyltin, retinoids, or fatty acids. I find that the argument of structural similarity between putative ligands should be used with caution and this should be clarified in the text.

Line 415: “or to modulate”

Reviewer 3 Report

The review addresses a specific issue on which there is controversy, the relationship between melatonin and the retinoic acid-related orphan receptor (ROR), and discusses whether ROR is a nuclear receptor of melatonin or if ROR is linked to melatonin in terms of its expression, function, and mechanism of action.

Throughout the manuscript, specific and important aspects indicating if the functions of ROR are related to or independent of melatonin are reviewed. Authors highlight that although various studies have reported that RORs mediate functions of melatonin, and that melatonin can regulate the transcriptional activity of RORs, it has also been found to function independently from melatonin-induced signalling.

No consensus has been reached on whether ROR is a nuclear receptor of melatonin. Authors conclude that the mechanisms underlying the effects exerted by melatonin need to be explored and that great care must be taken with studies that explore ROR and melatonin complex interactions.

The bibliography is very complete and the most interesting articles on the subject are cited.

Minor comments:

Line 41. Must be “N-Acetyl-5-methoxytryptamine” instead of “N-Acetyl-5-methoxytrypamine”.

In section 2.2. Functions of retinoic acid-related orphan receptor: related to or independent of melatonin interesting results such as that melatonin suppresses activation of hepatic stellate cells through RORα-mediated inhibition of 5-lipoxygenas, should be included (Shajari et al., J Pineal Res 2015).

Lines 177-180. “In addition to the immune system, the melatonin-RORγ axis also suppresses cancer [60]. In vitro and in vivo studies have shown that melatonin may induce growth and angiogenesis suppression in gastric cancer cells by down-regulating RORγ expression [60]”. Reference 60 (Chang et al., Molecular pharmacology 2015) does not correspond to the information provided in this paragraph.

Taking into account that the main objective of the review is to know the complex relationships between ROR and melatonin as well as to determine whether or not the hormone is an ROR ligand, it would be more appropriate a table with the main results for and against such interaction instead of Table 1, which merely summarizes the main natural and synthetic ROR ligands.

In References section a high percentage of bibliographic citations should be abbreviated (3,4,6,7,9,13,14,15,16,17,19,21,22,23,25,26,28,29,30,32,33,42,44,47,51,55,57,60,63,64,65.66,70,72,73,81,83,86,87,88,89,90,91,93,94,95,97,99,102,103,105,107,108,110,111,116,118,119,120,122,127,128,129).
